# Neural Combinatorial Optimization on Heterogeneous Graphs
# An Application to the Picker Routing Problem in Mixed-shelves Warehouses

**Primary Keywords:** *(2) Learning;*

## Abstract

In recent years, machine learning (ML) models capable of solving combinatorial optimization (CO) problems have received a surge of attention. While early approaches failed to outperform traditional CO solvers, the gap between hand-crafted and learned heuristics has been steadily closing. However, most work in this area has focused on simple CO problems to benchmark new models and algorithms, leaving a gap in the development of methods specifically designed to handle more involved problems. Therefore, this work considers the problem of picker routing in the context of mixed-shelves warehouses, which involves not only a heterogeneous graph representation, but also a combinatorial action space resulting from the integrated selection and routing decisions to be made. We propose both a novel encoder to effectively learn representations of the heterogeneous graph and a hierarchical decoding scheme that exploits the combinatorial structure of the action space. The efficacy of the developed methods is demonstrated through a comprehensive comparison with established architectures as well as exact and heuristic solvers.

## Introduction

In recent years, there has been a remarkable surge in the adoption of end-to-end ML methods for solving NP-hard CO problems, which offers an appealing alternative to the tedious task of manually crafting heuristics. Typically, neural combinatorial optimization (NCO) integrates deep neural networks as feature extractors in the reinforcement learning (RL) framework to enable a neural agent to learn optimal decisions based on the extracted information. Significant progress has been made in both of these domains, particularly with the Attention Model (AM) proposed by Kool, van Hoof, and Welling (2019) and improved training algorithms like POMO (Kwon et al. 2020), which led to substantial performance improvements. Nevertheless, most of these advances are designed for and applicable to only a limited number of simple CO problems, with the Travelling Salesman Problem (TSP) and the Vehicle Routing Problem (VRP) being the most frequently used problems (Mazyavkina et al. 2021). To foster its practical relevance, NCO also has to demonstrate its effectiveness in dealing with more complex real-world problems, which necessitates the design of novel models and algorithms. Therefore, in this work, we will focus on the Mixed-shelves Picker Routing Problem (MSPRP) as an example application to develop new methods within the NCO framework that are capable of addressing more complicated CO problems.

In a warehouse context, picker routing is concerned with generating efficient sequences of storage locations to be visited by a picker to collect items requested by customer orders and deliver them to a central packing station. Recent research has focused on picker routing in mixed-shelves warehouses, where items of the same stock keeping unit (SKU) are spread across multiple shelves, allowing pickers to choose between alternative storage locations and thus reduce the travel distances (Xie, Li, and Luttmann 2023). The integration of selection and routing decisions in MSPRP results in a complex combinatorial action space when formulating the problem as a sequential decision problem. Thus, we utilize the MSPRP as an example of a more complex CO problem to pioneer the development of neural architectures capable of addressing a broader spectrum of CO problems than those currently covered in existing NCO literature.

While Cals et al. (2021) are the first to apply RL to improve warehouse operations, they focus on the problem of order batching. To the best of our knowledge, no work has yet used this framework to generate solutions for the picker routing problem. Therefore, we first formulate the MSPRP as a Markov Decision Process (MDP), where we represent the state of the problem instance as a heterogeneous graph. Further, we develop a novel encoder-decoder architecture, where the encoder is specifically designed to effectively learn representations over the heterogeneous graph structure. The decoder then uses these representations to select actions from a factorized action space in a hierarchical manner. To the best of our knowledge, this work is the first in the NCO field to perform an action space factorization using a hierarchical decoding strategy. Other work dealing with multidimensional action spaces either flatten it by including every possible combination of the distinct action space dimensions or solve only parts of the decision problem using a neural agent and others by handcrafted heuristics. We will demonstrate that both our encoder and decoder models perform better than the other methods currently employed in the literature. Therefore, this work not only contributes to the field of NCO by demonstrating its effectiveness on a novel problem class, but also by developing new models that hopefully allow more CO problems with similar problem structure to be effectively solved using NCO.

## Related Work

**Mixed-shelves Picker Routing.** The picking process is known to be a very labor-intensive and time-consuming task, representing an estimated 50-65% of the total operating cost for a warehouse (De Koster, Le-Duc, and Roodbergen 2007). Within conventional picker-to-parts warehouses, the majority of a pickers' working time is dedicated to traveling between shelves (Tompkins 2010). Minimizing the travel distance of order pickers has consequently been extensively studied in the operations research community and a vast body of literature on the picker routing problem has emerged over the years. We refer to the literature review of De Koster, Le-Duc, and Roodbergen (2007) for a comprehensive overview on picker routing and related warehousing decision problems. Here we focus in particular on picker routing in mixed-shelves warehouses which can be regarded a combined problem of selecting the storage positions to visit and the actual picker routing. The first to treat this kind of problem were Daniels, Rummel, and Schantz (1998) who considered a TSP variant of the MSPRP, where the set of orders to be collected during a tour is assumed to be determined by a preceding order batching step. The authors define a mathematical model for the problem and propose a tabu search algorithm to solve it. Later, Weidinger (2018) solve the same problem under the assumption of a rectangular warehouse using a more efficient heuristic based on the dynamic programming algorithm of Ratliff and Rosenthal (1983). More recently, Weidinger, Boysen, and Schneider (2019) and Xie, Li, and Luttmann (2023) extend the MSPRP by integrating the decision which orders to pick during a tour into the problem. Their problem can be regarded a variant of the capacitated VRP due to the construction of multiple tours with limited capacity. While Weidinger, Boysen, and Schneider (2019) propose different construction heuristics and an iterative improvement procedure based on destroy and repair operators, Xie, Li, and Luttmann (2023) develop a Variable Neighborhood Search (VNS) to solve this problem.

**Neural Combinatorial Optimization.** The Ptr-Net proposed by Vinyals, Fortunato, and Jaitly (2015) was the first architecture that was specifically designed to solve CO problems. The authors used supervised learning to train the network to solve the TSP using optimal solutions as training data. Their architecture follows an encoder-decoder structure, where the encoder maps customer locations from the feature space to learned embeddings, which in turn are used by the decoder to iteratively select the next location to visit. Later, Bello et al. (2017) proposed a Deep Reinforcement Learning (DRL) approach to train Ptr-Nets without the need for a database of optimal solutions. Network architectures other than the Recurrent Neural Networks used in the Ptr-Net were proposed by Nazari et al. (2018) and Kool, van Hoof, and Welling (2019). The latter introduced the AM which incorporates the multi-head attention mechanism of Vaswani et al. (2017) in both encoder and decoder of the Ptr-Net and is still the state-of-the-art NCO architecture.

Recently, Mazyavkina et al. (2021) provided a structured literature analysis of RL for CO problems. The authors found that the majority of the literature proposes methods for simple CO problems, with the TSP and capacitated VRP being covered the most. A notable exception is the work of Kwon et al. (2021), who propose a novel encoder based on the AM that is applicable to a wider range of CO problems that can be expressed as bipartite graphs.

On the other hand, new decoder models for more complex CO problems have hardly been investigated, as also Hildebrandt, Thomas, and Ulmer (2022) note. They identified that most research applying RL to more complex VRP variants uses either an action space restriction or a simplification thereof. In the former case, the RL agent is only used to generate solutions for a sub-problem, and everything else is solved using established heuristics. An example is Chen, Ulmer, and Thomas (2022), who also face a combined selection and routing problem where they use a neural agent to select a vehicle from a heterogeneous fleet, but rely on handcrafted heuristics for route generation. An example of action space simplification can be found in Song et al. (2022), who propose a heterogeneous graph attention network as a feature extractor for the job-shop scheduling problem, where jobs and machines pose different node types to be represented by the encoder. In the end, they propose to concatenate the learned embeddings of jobs and machines and consider a combinatorial actions space with all job-machine combinations in the decoding stage. Although this flattening of the action space is straightforward to implement, an obvious drawback is the size of the action space growing quadratically with the number of jobs and machines.

## Formal Definition of the MSPRP

This work considers an MSPRP similar to the one outlined by Weidinger, Boysen, and Schneider (2019). However, Xie et al. (2021) proposed the concept of *split orders*, which allows items of an order to be picked within different tours, resulting in significantly shorter picker routes. Inspired by their findings, we relax the constraint that items of the same order must be collected in one picker tour. This adjustment is further justified by our focus on single-depot instances, ensuring that items from a single order reach the same packing stations for consolidation and packaging, regardless of the tour in which they were picked.[1]

The goal of our MSPRP is then to pick all $d_p$ demanded units of all requested SKUs $p \in \mathcal{P}$ while minimizing the travel distance of the various picker tours $b \in \mathcal{B}$. A tour is defined by the storage locations visited between two successive visits to the packing station $h$, where picked items are unloaded and commissioned. During a tour, no more than $\kappa$ units can be picked. We assume that the demand for an SKU must not be satisfied by a single picker tour, but can be split over multiple tours and collected from different shelves (*split deliveries*). Each SKU may be retrieved from multiple shelves / racks $i \in \mathcal{V}^R$ of the warehouse. However, for the formulation of the mathematical model, we follow the approach of Weidinger, Boysen, and Schneider (2019) and

---

[1]The MSPRP covered here can be considered a special case of the problem described by Weidinger, Boysen, and Schneider (2019), where each order consists of only a single SKU. The authors provide a proof for the NP-hardness of the MSPRP.

consider storage locations $i \in \mathcal{V}^{S}$ instead of shelves. Accordingly, each shelf consists of multiple storage locations or compartments, each of which can store units of only a single SKU. The distance $D_{ij}$ between storage locations of the same shelf is zero. To collect an SKU $p$ during a tour, a picker can choose between the different storage locations $i \in \mathcal{V}_p^{S}$ storing $n_i > 0$ units of this SKU. Further, to formulate the mathematical model, we need to determine the number of tours required to fully satisfy the demand. Due to the assumption of split orders and split deliveries, this is simply the ceiling of the total demand divided by the picker capacity: $|\mathcal{B}| = \left\lceil \frac{\sum_p d_p}{\kappa} \right\rceil$. Table 1 summarizes the notation used to define the mathematical model.

$$\text{Min} \quad Z = \sum_{(i,j) \in \mathcal{E}} \sum_{b \in \mathcal{B}} D_{ij} \cdot x_{ijb} \tag{1}$$

s.t.

$$\sum_{(i,j) \in \mathcal{E}} x_{ijb} = \sum_{(j,i) \in \mathcal{E}} x_{jib} \quad \forall\, i \in \mathcal{V}^{L}, b \in \mathcal{B} \tag{2}$$

$$\sum_{(i,j) \in \mathcal{E}} x_{ijb} \leq 1 \quad \forall\, i \in \mathcal{V}^{L}, b \in \mathcal{B} \tag{3}$$

$$M \cdot \sum_{i \in \mathcal{V}^{L}} x_{ijb} \geq y_{jb} \quad \forall\, j \in \mathcal{V}^{S}, b \in \mathcal{B} \tag{4}$$

$$\sum_{j \in \mathcal{V}^{S}} x_{hjb} = 1 \quad \forall\, b \in \mathcal{B} \tag{5}$$

$$\sum_{i \in S} \sum_{j \in S} x_{ijb} \leq |S| - 1 \quad \forall\, b \in \mathcal{B},\, S \subset \mathcal{V}^{S}, |S| \geq 2 \tag{6}$$

$$\sum_{i \in \mathcal{V}^{S}} y_{ib} \leq \kappa \quad \forall b \in \mathcal{B} \tag{7}$$

$$\sum_{i \in \mathcal{V}_p^{S}} \sum_{b \in \mathcal{B}} y_{ib} = d_p \quad \forall\, p \in \mathcal{P} \tag{8}$$

$$\sum_{b \in \mathcal{B}} y_{ib} \leq n_i \quad \forall\, i \in \mathcal{V}^{S} \tag{9}$$

$$x_{ijb} \in \{0, 1\} \quad \forall\, (i,j) \in \mathcal{E}, b \in \mathcal{B} \tag{10}$$

$$y_{ib} \geq 0 \quad \forall\, i \in \mathcal{V}^{S}, b \in \mathcal{B} \tag{11}$$

The objective function (1) minimizes the total distance travelled by the picker. Every storage location that is visited during a picker tour must also be left, which is ensured by constraints (2). Moreover, constraints (3) ensure that a storage location is visited only once during a tour. Although a storage location may be visited multiple times if the remaining capacity of the picker does not suffice to satisfy the respective demand in one go, it does not make sense to visit the same storage location twice within one tour.

Through the Big-M formulation in (4) we make sure that only items are taken from a storage location if this location is also included in the respective tour. As no more than $\kappa$ items may be picked in a single tour, it is sufficient to set $M = \kappa$. Since every tour must start and end at the depot, equations (5) require that the depot is left exactly once in

| | |
|---|---|
| $\mathcal{P}$ | Set of SKUs for picking |
| $\mathcal{V}^{L}$ | Set of storage locations and depot ($\mathcal{V}^{L} = \mathcal{V}^{S} \cup \{h\}$) |
| $\mathcal{E}$ | Set of edges $\{(i,j) \mid i, j \in \mathcal{V}^{L}, i \neq j\}$ |
| $\mathcal{V}_p^{S}$ | Set of storage locations including picking item $p \in \mathcal{P}$ |
| $\mathcal{B}$ | Set of required tours $\{1, 2, ..., |\mathcal{B}|\}$ |
| $D_{ij}$ | Distance between two nodes $(i,j) \in \mathcal{E}$ |
| $\kappa$ | Maximum picking capacity per tour |
| $d_p$ | Total demand for item $p \in \mathcal{P}$ |
| $n_i$ | Available supply at storage location $i \in \mathcal{V}^{S}$ |
| $x_{ijt}$ | Binaray variable, indicating whether Node $j \in \mathcal{V}^{L}$ has been visited after node $i \in \mathcal{V}^{L}$ in tour $b \in \mathcal{B}$ |
| $y_{it}$ | Units picked up at location $i \in \mathcal{V}^{S}$ in tour $b \in \mathcal{B}$ |

Table 1: Notation used in the MIP-Model

every tour. In combination with the network flow constraints (2) this also ensures that each tour goes back to the depot. And the subtour elimination constraints (6) ensure that all storage locations visited in between are also connected to the tour. Constraints (7) ensure that the picker capacity is not exceeded. Furthermore, constraints (8) make sure that customer orders are satisfied and inequalities (9) take care that the available stock of items in a shelf is not exceeded during order picking. Lastly, constraints (10) and (11) delineate the domains of the decision variables $x$ and $y$.

## Deep Reinforcement Learning for the MSPRP

Similar to other CO problems, a solution to the MSPRP can be obtained via a sequence of decisions, which can naturally be constructed using RL. In each iteration $t = 0, \ldots, t_{\max}$, the RL agent decides the next location to visit and which items to pick until the demand for all SKUs is met. We model the solution construction as an MDP with the policy being a $\theta$-parameterized neural network $\pi_\theta$ which takes the state representation as input and outputs a probability distribution over the action space. In this chapter, we will first define the state, action space, transition rule and reward function of the MSPRP. Then, our policy network based on the encoder-decoder framework will be described in detail.

## MDP for the heterogeneous Graph MSPRP

**State.** Typically, the state of a CO problem is represented by a graph. For the MSPRP we found two possible graph representations. First, one could consider each available shelf and SKU combination as a unique storage location, similar to our mathematical model. However, a challenge arises in finding a size-agnostic feature representation for storage locations, which needs to include the supply and demand of the SKU contained in the storage location. Given that an SKU can exist in multiple locations, a representation ensuring unique identification of the stored SKU is necessary. The mathematical model addresses this by defining a subset $\mathcal{V}_p^{S}$ of locations containing SKU $p$. However, the encoder requires a vectorized representation, for instance, using a one-hot representation. Yet, this approach violates the requirement of the model being agnostic to the instance size, an important generalization property of NCO models.

Thus, a more natural way of representing the state of an MSPRP instance is by using a heterogeneous graph with different node types, namely shelves and SKUs. Let at each decision step $t$, the state $s_t \in \mathcal{S}$ be a graph $\mathcal{G} = (\mathcal{V}, \mathcal{P}, E_t, c_t)$ representing the current status of the problem. The set of location nodes $\mathcal{V}$ is the union between the packing station $h$ and the shelves $\mathcal{V}^{\mathrm{R}}$. We follow the common practice in the NCO field and represent the location nodes by their Cartesian coordinates $\mathbf{x}_i^{\mathcal{V}} \in \mathbb{R}^2$. The set of SKUs $\mathcal{P}$ poses the second type of nodes in the heterogeneous graph. We use $\mathbf{x}_p^{\mathcal{P}}$ to denote the feature representation of an SKU $p$, which consists only of its demand $d_{p,t}$ at time $t$. Moreover, edges with weights $E_t$ connect shelf and SKU nodes, specifying the storage quantity $e_{ip,t}$ of an item $p$ in the respective shelf $i$ at time $t$. Lastly, the graph is augmented by the context node $c_t$, containing dynamic context information relevant for the decoding stage. It consists of the current location $i_t$ of the picker as well as the remaining picker capacity $\kappa_t$.

**Action.** An action $\mathbf{a}_t \in \mathcal{A}(s_t)$ corresponds to a feasible shelf-SKU combination $(i, p)$ specifying the next picking job. Given $s_t$, visiting shelf $i$ is a feasible action if it can satisfy the demand of at least one SKU currently in demand. Furthermore, given the next picking location $i$, an SKU $p$ may only be selected if it is available in the respective shelf. The depot can always be visited to replenish the picker's capacity, but consecutive visits to the depot are prohibited. When the picker's capacity is exhausted, visiting the depot is the only possible action. We introduce a dummy SKU $|\mathcal{P}|+1$ that is selected when the depot is visited. Furthermore, we assume that the picker always starts and ends the tour at the depot, i.e. $\mathbf{a}_0 = \mathbf{a}_{t_{\max}} = (h, |\mathcal{P}| + 1)$.

**Transition.** Once, a feasible shelf-SKU pair $(i, p)$ has been selected, the transition function $\Gamma_\bullet(s_t, \mathbf{a}_t)$ deterministically determines the next state $s_{t+1}$. To this end, the picking quantity $y_t$ is determined in a first step. Here, we choose to always pick the maximum feasible quantity, that is the minimum of the shelf's supply for the SKU, the SKU's demand and the remaining picker capacity. Given $\mathbf{a}_t$ and $y_t$, the environment deterministically transits to a new state $s_{t+1}$. It consists of the new location $i_t \in \mathcal{V}$, the updated demand $d_{p,t+1} = d_{p,t} - y_t$, the updated supply $e_{ip,t+1} = e_{ip,t} - y_t$ and the remaining picker load $\kappa_{t+1} = \kappa_t - y_t$. When the agent chooses to return to the depot, the picker capacity will be set to the initial payload $\kappa$ and supply and demand remain unchanged. The problem instance is solved once the demand for every SKU is met and the picker has returned to the depot. We use $\boldsymbol{\tau}$ to denote the complete solution starting and ending at the depot, i.e. $\boldsymbol{\tau} = \{\mathbf{a}_t, t = 0, ..., t_{\max}\}$.

**Reward.** To minimize the total travel distance of the picker, we define the reward $R(\boldsymbol{\tau})$ as the negative of the objective value defined in equation (1), i.e. the summation of distances between locations visited in $\boldsymbol{\tau}$.

## Heterogeneous Graph Neural Network

**Encoder.** In this work we are facing a CO problem which is defined over a heterogeneous graph with different node types, namely shelves and SKUs. Heterogeneous nodes have different properties, manifested in distinct feature spaces. This requires specialized architectures specifically designed for handling heterogeneous inputs and relationships between them. Several models have been proposed in the literature for learning representations over heterogeneous graphs. The Heterogeneous Graph Attention (HAN) architecture, introduced by Wang et al. (2019), adapts the Graph Attention Network (GAT) to handle graphs with diverse node types. To this end, the authors propose type-specific transformations to project the different node-types from their distinct feature spaces into a mutual embedding space of dimensionality $d_h$. The mutual embedding space is then used to calculate attention scores $\alpha$ between the different nodes:

$$\alpha_{ij}^l = \frac{\exp(\epsilon_{ij}^l)}{\sum_{q \in \mathcal{N}_i} \exp(\epsilon_{iq}^l)}, \quad \epsilon_{ij}^l = \sigma\left(\mathbf{w}^{l\top}[\mathbf{h}_i^{l-1} || \mathbf{h}_j^{l-1}]\right) \quad (12)$$

where, $\mathbf{w}^l \in \mathbb{R}^{2d_h}$ is a learnable weight vector, $[\cdot || \cdot]$ denotes vertical concatenation, $\sigma$ represents a non-linear activation function, and $\mathcal{N}_i$ is the neighborhood of node $i$, including node $i$ itself. Moreover, $\mathbf{h}_i^l$ denotes the embedding of node $i$ after the $l$th layer, where $\mathbf{h}_i^0$ is the initial embedding obtained via the type-specific transformations $W_{\phi_i}$ for a node of type $\phi_i$. The attention scores are then used to compute the node embeddings $\mathbf{h}_i^l$ as a weighted sum over all embeddings $\mathbf{h}_j^{l-1}$ of nodes $j$ in the neighborhood of $i$, along with the embedding $\mathbf{h}_i^{l-1}$ itself. One drawback of this architecture is that no edge weights are utilized when computing the attention score. Therefore, the HGCN proposed by Yang et al. (2021) adapts the Graph Convolution Network (Kipf and Welling 2017) to aggregate the node embeddings from the neighborhood of a node using the adjacency matrix directly. HGCN then uses an attention mechanism similar to HAN to compute a weighted sum of the target node embedding and the aggregation of the neighborhood embeddings.

For CO problems formulated over homogeneous graphs like the TSP, the AM has been shown to outperform other graph neural networks such as the GNN proposed by Khalil et al. (2017) by a large margin. Therefore, Kwon et al. (2021) adapt the attention model to solve CO problems defined over bipartite graphs with node types $\mathcal{I}$ and $\mathcal{J}$ as well as weights $E \in \mathbb{R}^{|\mathcal{I}| \times |\mathcal{J}|}$ of the edges connecting nodes from the two sets. The encoder of their MatNet architecture also uses type-specific transformations to generate initial embeddings $\mathbf{h}_i^0$ of size $d_h$ for all nodes. Then, for each of the node types the authors apply distinct update functions $\mathcal{F}_\mathcal{I}$ and $\mathcal{F}_\mathcal{J}$. The update functions perform $L$ layers of multi-head cross-attention to calculate attention scores between a target node $i$ and all nodes $j$ from the respective other set. To this end, let $\mathcal{Z}$ be the set of all nodes $i \in \mathcal{I} \cup \mathcal{J}$, $\mathcal{Z}_{\phi_i}$ the subset of nodes of the same type as $i$ and $\mathcal{Z}_{\phi_i}^{\complement}$ the set of nodes of the respective other type. Then, cross-attention is defined as:[2]

$$\alpha_{ij}' = \frac{\mathbf{q}_i^\top \mathbf{k}_j}{\sqrt{d_k}}, \qquad \forall i \in \mathcal{Z}, \, j \in \mathcal{Z}_{\phi_i}^{\complement} \qquad (13)$$

---

[2]for succinctness we omit head and layer enumeration

where

$$\mathbf{q}_i = W_{\phi_i}^Q \mathbf{h}_i^{l-1} \qquad \mathbf{k}_j = W_{\phi_i}^K \mathbf{h}_j^{l-1} \qquad (14)$$

and weight matrices $W_{\phi_i}^Q$ and $W_{\phi_i}^K \in \mathbb{R}^{d_k \times d_h}$ being learned by the update function corresponding to nodes of type $\phi_i$. After that, Kwon et al. (2021) propose to concatenate $\alpha'_{ij}$ with the corresponding edge weight $e_{ij}$ and map it through a feed-forward neural network FF : $\mathbb{R}^2 \rightarrow \mathbb{R}$ to a scalar score, which is then normalized using the softmax function:

$$\alpha_{ij} = \frac{\exp(\epsilon_{ij})}{\sum\limits_{q \in \mathcal{Z}_{\phi_i}^{\mathsf{C}}} \exp(\epsilon_{iq})}, \quad \epsilon_{ij} = \mathrm{FF}\big([\alpha'_{ij}||e_{ij}]\big) \qquad (15)$$

The resulting weights are used to compute a weighted average of the embeddings $\mathbf{v}_j = W_{\phi_i}^V \mathbf{h}_j^{l-1}$ of the nodes in $\mathcal{Z}_{\phi_i}^{\mathsf{C}}$. In the end, skip connections, layer normalization (LN) and feed-forward layers are used as in Vaswani et al. (2017):

$$\hat{\mathbf{h}}_i^l = \mathrm{LN}\left(\mathbf{h}_i^{l-1} + \sum_{j \in \mathcal{Z}_{\phi_i}^{\mathsf{C}}} \alpha_{ij} \mathbf{v}_j\right), \qquad \forall i \in \mathcal{Z} \qquad (16)$$

$$\mathbf{h}_i^l = \mathrm{LN}\left(\hat{\mathbf{h}}_i^l + \mathrm{FF}^l(\hat{\mathbf{h}}_i^l)\right), \qquad \forall i \in \mathcal{Z} \qquad (17)$$

Although MatNet poses a viable extension of the AM, we consider the limitation to bipartite graphs as a shortcoming. In bipartite graphs, relations exist only between nodes of different types, yet many real-world problems formulated over heterogeneous graphs also involve intra-type relationships. For instance, in the MSPRP shelf nodes strongly influence each other due to the potential availability of the same SKU in multiple shelves. By exclusively focusing on inter-type relationships through the application of cross-attention, the MatNet encoder may not fully encompass such relationships in the node embeddings.

Motivated by the identified weaknesses of existing approaches, we propose the Heterogeneous Attention Model (HAM), which is an extension of the MatNet encoder for the more general case of a heterogeneous graph. As usual, our HAM encoder first projects the feature vectors into a mutual embedding space using type-specific transformations. Then, multi-head self-attention (MHSA), followed by skip-connection and layer normalization, as described by Vaswani et al. (2017), is performed for each node type:

$$\tilde{\mathbf{h}}_i^l = \mathrm{LN}(\mathbf{h}_i^{l-1} + \mathrm{MHSA}_{\phi_i}^l(\{\mathbf{h}_j^{l-1} \,|\, j \in \mathcal{Z}_{\phi_i}\})), \quad \forall i \in \mathcal{Z}$$

Using self-attention allows the network to learn relationships between nodes of the same type, like the substitutability of shelves containing the same SKUs.

Next, we apply the operations described in equations (13-17) to capture inter-type relationships, but we replace $\mathbf{h}^{l-1}$ in equations (14) and (16) with $\tilde{\mathbf{h}}^l$. Moreover, we remove the feed-forward layer and the concatenation in equation (15) and simply multiply $\alpha'_{ij}$ and $e_{ij}$, which yielded better results in our experiments while being much faster. Figure 1 depicts the architecture of our HAM encoder.

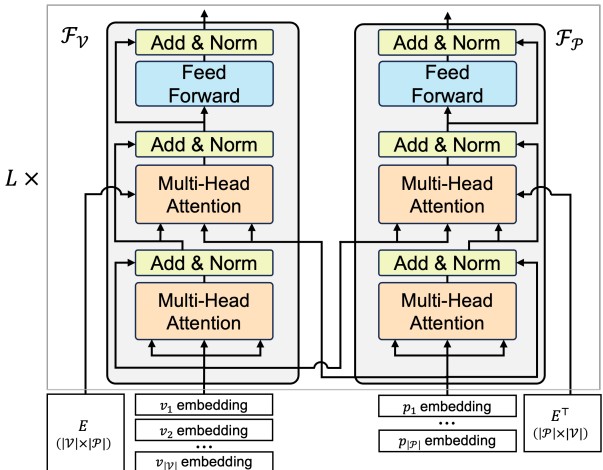

Figure 1: HAM Encoder Architecture

**Decoder.** This work uses a hierarchical decoder to sample actions specifying the next location $i_t$ to visit and the SKU $p_t$ to pick. To this end, we decompose the action space of composite actions $\mathcal{A}$ into the sub-action spaces $\mathcal{A}^\mathcal{V} \equiv \mathcal{V}$ and $\mathcal{A}^\mathcal{P} \equiv \mathcal{P}$, where $\mathcal{A} = \mathcal{A}^\mathcal{V} \times \mathcal{A}^\mathcal{P}$. Similar to Sharma et al. (2017), our hierarchical decoding strategy uses a separate policy network for each sub-action space. However, in contrast to Sharma et al. (2017), we do not sample from the shelf- and SKU-policy independently, but in a hierarchical manner. Therefore, let $\Gamma_\circ : \mathcal{S} \times \mathcal{A}^\mathcal{V} \rightarrow \mathcal{S}'$ be a partial transition function generating an intermediate state $s_t'$ with updated location information $i_t$. Moreover, let $\pi_{\theta_\mathcal{V}} : \mathcal{S} \rightarrow \mathcal{A}^\mathcal{V}$ be the shelf-policy and $\pi_{\theta_\mathcal{P}} : \mathcal{S}' \rightarrow \mathcal{A}^\mathcal{P}$ the SKU-policy, then sampling a composite action $\mathbf{a}$ from $\pi_\theta(\mathbf{a}_t \,|\, s_t)$ is equivalent to sampling shelves and SKUs successively from $\pi_{\theta_\mathcal{V}}$ and $\pi_{\theta_\mathcal{P}}$, respectively, according to the chain rule of probability:

$$\pi_\theta(\mathbf{a}_t \,|\, s_t) = \pi_{\theta_\mathcal{V}}(i_t \,|\, s_t) \cdot \pi_{\theta_\mathcal{P}}(p_t \,|\, s_t') \qquad (18)$$

While the sub-policies share the same encoder they have their own decoder networks. We use the decoder architecture from the AM for both the shelf- and the SKU-policy. The attention decoder for sub-action space $\mathcal{A}_d$ performs multi-head attention by using the embeddings $\mathbf{h}_i^L$ of the respective node type from the encoder to compute keys and values. Moreover, the context node $c_t$ is used by each decoder to compute the query (again omitting head enumeration):

$$\mathbf{q}_c = W_d^Q \mathbf{h}_c \quad \mathbf{k}_i = W_d^K \mathbf{h}_i^L \quad \mathbf{v}_i = W_d^V \mathbf{h}_i^L \quad \forall i \in \mathcal{A}_d$$

where $\mathbf{h}_c$ is the concatenation of the encoder embedding of the current picker location $\mathbf{h}_{i_t}^L$ and the remaining picker capacity $\kappa_t$. Again, $W_d^Q$, $W_d^K$ and $W_d^V$ are weight matrices learned per head and sub-action space $\mathcal{A}_d$. The MHSA operation generates a glimpse, similar to Bello et al. (2017). The glimpse is multiplied with a single-head transformation of $\mathbf{h}_i^L$ to obtain the logit for action $i$ in the respective sub-space. Lastly, softmax-normalization of the logits over all actions in the sub-space yields their sampling probabilities.

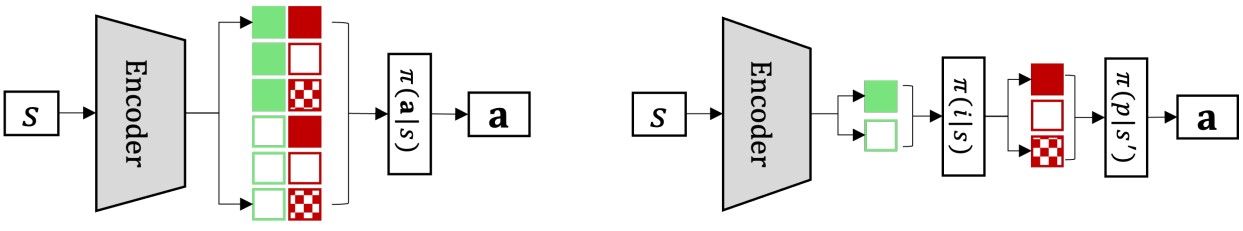

(a) Policy $\pi$ on combinatorial action space      (b) Sub-policies on factored action space

Figure 2: Comparison of policy networks in combinatorial and factored action spaces

Figure 2 illustrates our hierarchical decoder and compares it to a policy operating on the composite action space as implemented by Song et al. (2022) for instance.

**Training.** To learn the parameters $\theta$ of the policy network, we adapt the POMO training algorithm proposed by Kwon et al. (2020). POMO is based on the well-known REINFORCE gradient estimator (Williams 1992):

$$\nabla_\theta \mathcal{L}(\theta) \approx \frac{1}{B} \sum_{i=1}^{B} \left( R(\tau_i) - b(s_i) \right) \nabla_\theta \log \pi_\theta(\tau_i | s_i), \quad (19)$$

where $\pi_\theta(\tau_i | s_i) = \prod_{t=2}^{t_{\max}} \pi_\theta(\mathbf{a}_{i,t} | s_{i,t})$, $b$ is a baseline function, used to reduce the variance of the estimated gradients and $B$ is the size of the mini-batch. In POMO, the baseline for a training instance is computed as the average reward of $W$ sampled solutions for that instance. To generate diverse sampled solutions, POMO employs $W$ distinct initial actions, from which the policy network constructs trajectories through Monte-Carlo rollouts. For the MSPRP this means using $W$ different shelf-SKU combinations for the first picking task $\mathbf{a}_1$. However, unlike TSP or CVRP where all locations need to be visited anyway, the MSPRP does not require all shelves to be visited. Consequently, selecting a random shelf at the start of the picker tour can drastically deteriorate the solution quality, when this shelf is at the other end of the warehouse. Consequently, selecting the initial actions randomly may result in a high variance of the baseline, especially for small $W$. Therefore, instead of selecting $W$ random actions, we sample them from the policy $\pi_\theta(\mathbf{a}|s)$ without replacement (wor). To ensure predominantly favorable initial moves right from the start of the training process, we warm the policy network up for one epoch using an exponential moving average over the mini-batch rewards as baseline, similar to Kool, van Hoof, and Welling (2019). Given the baseline $b(s_i)$, gradients are computed using equation (19) and we use the Adam optimizer (Kingma and Ba 2015) with a learning rate of 2e-4 to obtain the parameter updates. Algorithm 1 summarizes this procedure which we will refer to as Policy-aware POMO (Pa-POMO). Empirically, we found that, using the Pa-POMO training algorithm, the variance of the baseline value is reduced by 20% compared to POMO. Figure 3 shows how the variance of the gradient updates for POMO and Pa-POMO evolve during training, demonstrating the effectiveness of our Pa-POMO method for the MSPRP.

---

**Algorithm 1: Pa-POMO Training**

---

**Input**: training set $S$, POMO size $W$, batch size $B$, policy $\pi_\theta$, number of training steps $T$

1: Warm up the policy using exp. moving average baseline
2: **for** $step = 1, ..., T$ **do**
3:     $s_1, \ldots, s_B \overset{iid}{\sim} S$
4:     $\{\mathbf{a}_{i,1}^1, \mathbf{a}_{i,1}^2, \ldots, \mathbf{a}_{i,1}^W\} \overset{wor}{\sim} \pi(\mathbf{a}_i | s_i) \quad \forall i = 1, ..., B$
5:     $s_i^j \leftarrow \Gamma_\bullet(s_i, \mathbf{a}_{i,1}^j) \quad \forall i = 1, ..., B, \, j = 1, ..., W$
6:     $\tau_i^j \leftarrow \pi_\theta(\tau_i^j | s_i^j) \quad \forall i = 1, ..., B, \, j = 1, ..., W$
7:     $b_i = \frac{1}{W} \sum_{j=1}^{W} R(\tau_i^j) \quad \forall i = 1, ..., B$
8:     $A_i^j \leftarrow \left( R(\tau_i^j) - b_i \right) \quad \forall i = 1, ..., B, \, j = 1, ..., W$
9:     $\nabla_\theta \mathcal{L} \leftarrow \frac{1}{B \cdot W} \sum_{i=1}^{B} \sum_{j=1}^{W} A_i^j \nabla_\theta \log \pi_\theta(\tau_i^j | s_i^j)$
10:    $\theta \leftarrow \text{Adam}(\theta, \nabla_\theta \mathcal{L})$
11: **end for**

---

## Experiments

### Experimental Setup and Instance Generation

**Dataset.** For training and evaluating the proposed policies, we generate instances for three warehouse types that differ in the number of available storage locations and shelves. In conformity with other NCO literature, that typically tests models on TSP and VRP instances with 20, 50 and 100 nodes, we consider instances with 20, 50 and 100 storage locations $\mathcal{V}^S$ referred to as PRP20, PRP50 and PRP100, respectively. While the number of shelves is fixed to 10, 25 and 40 for the respective instances, we alter the number of demanded SKUs for each warehouse type.

We follow the standard procedure in NCO literature and sample the $|\mathcal{V}^R|$ shelf locations uniformly at random in the unit square and assume Euclidean distances between them. We choose this approach over a realistic rectangular warehouse layout with parallel racks and non-Euclidean distance as described for example in Weidinger (2018), as this would introduce additional complexity through the requirement of a distance matrix. Since we are more interested in demonstrating the ability of the proposed methods to solve CO problems over heterogeneous graphs, we sacrifice some realism here to facilitate model development.

We randomly select the $|\mathcal{V}^S|$ storage locations from all $|\mathcal{P}| \times |\mathcal{V}^R|$ possible SKU-shelf combinations and sample the supply from a discrete uniform distribution with mean $\bar{n}_i$. Likewise, the demand for each SKU is sampled from a

| | PRP20 | | | PRP50 | | | PRP100 | |
|---|---|---|---|---|---|---|---|---|
| $|\mathcal{V}^{\mathrm{R}}|$ | — | 10 | — | — | 25 | — | – | 40 | – |
| $|\mathcal{V}^{\mathrm{S}}|$ | — | 20 | — | — | 50 | — | – | 100 | – |
| $\bar{d}_p$ | — | 2.5 | — | — | 2.5 | — | – | 2.5 | – |
| $|\mathcal{P}|$ | 3 | 6 | 9 | 12 | 15 | 18 | 15 | 20 | |
| $\bar{n}_i$ | 1 | 1.5 | 2 | 1 | 1.5 | 1.5 | 1 | 1 | |
| $\kappa$ | 6 | 9 | 9 | 12 | 12 | 15 | 12 | 15 | |

Table 2: Parameter values for instance generation

| | | Encoder | | | | |
|---|---|---|---|---|---|---|
| | $|\mathcal{P}|$ | AM | HAN | HGCN | MatNet | HAM |
| PRP20 | 3 | 1.543 | 1.636 | 1.589 | 1.491 | **1.476** |
| | 6 | 2.425 | 2.498 | 2.358 | 2.261 | **2.216** |
| | 9 | 2.932 | 2.976 | 2.898 | 2.782 | **2.727** |
| PRP50 | 12 | 3.980 | 4.044 | 3.968 | 3.654 | **3.538** |
| | 15 | 4.129 | 4.274 | 4.087 | 3.895 | **3.740** |
| | 18 | 4.266 | 4.363 | 4.261 | 3.959 | **3.860** |

Table 3: Comparison of different encoders

| | | Decoder | | |
|---|---|---|---|---|
| | $|\mathcal{P}|$ | Combinatorial | Hybrid | Hierarchical |
| PRP20 | 3 | 1.537 | 1.496 | **1.476** |
| | 6 | 2.342 | 2.245 | **2.216** |
| | 9 | 2.847 | 2.750 | **2.727** |
| PRP50 | 12 | 4.085 | 3.622 | **3.538** |
| | 15 | 4.284 | 3.840 | **3.740** |
| | 18 | 4.523 | 3.895 | **3.860** |

Table 4: Comparison of different decoders

discrete uniform distribution with mean $\bar{d}_p$. Lastly, we clip the demand of an SKU by the warehouse's total supply for it in order to ensure feasibility of all generated instances. Table 2 summarizes the parameters of the different instances.

**Setup.** We empirically validate the effectiveness of the proposed encoder-decoder architecture for solving the MSPRP in two steps. First, we evaluate the encoder as well as the hierarchical decoder of our HAM by comparing them with other approaches that we found in the literature. Then, we compare our DRL model with the solutions obtained by a heuristic and an exact solver.

For the first part, we use the HAN, HGCN and MatNet encoders we introduced earlier alongside our HAM encoder. Moreover, we use a homogeneous graph formulation of the MSPRP, using storage locations as sole node types, to solve it using the AM. For all encoder architectures, we are using our hierarchical decoder. One exception from this is the homogeneous graph formulation, which naturally has a flat action space and thus uses a single AM decoder. On the contrary, we validate the effectiveness of the hierarchical decoder by comparing it with a single AM decoder operating on the combinatorial action space similar to Song et al. (2022) as well as a hybrid policy, combining a neural agent and a handcrafted heuristic, like Chen, Ulmer, and Thomas (2022). This policy uses the AM decoder only to determine the next location to visit and, given a shelf, selects the SKU for which the most demand can be satisfied. Each of these decoders uses the HAM encoder as a feature extractor.

All policy networks are trained for 50 epochs on 350,000 training instances that are randomly generated on the fly, and evaluated on 30,000 test instances for each of the six instance types corresponding to layouts PRP20 and PRP50. We refrain from comparing all models on the PRP100 instances due to the immense computational training effort. These instances are only used in the second part of the experiments, where we compare our DRL model with a heuristic and the solutions obtained by the Gurobi solver, for which we set a time limit of 1 hour. For this part of the experiments we use the same training procedure but a separate test set of 20 instances.

**Model Configuration.** We use $L = 4$ layers for all encoders. The embedding dimension is set to $d_h = 256$ for the MatNet and HAM encoders and to $d_h = 512$ for the AM, HAN and HGCN encoders. For models relying on the multi-head attention mechanism, the number of heads is set to 8. During training, the batch size is set to 512 for all instances

of type PRP20 and PRP50 and to 256 for PRP100 instances. Moreover, we use $W = 10$ POMO samples. During testing, beam search (see e.g. Neubig (2017)) with a beam width of 100 is used. Models are trained and evaluated using a single Nvidia A100-80GB GPU and Gurobi as well as the heuristic are executed on an Intel Xeon E5-2690 v4 CPU. The code, test datasets and configuration files are publicly available.[3]

**Heuristic for comparison.** In order to validate the solution quality of our DRL method on instances where Gurobi cannot find the optimal solution within the time limit, we include the VNS of Xie, Li, and Luttmann (2023) into the comparison, where we can make some simplifications due to our split-order and single-depot assumptions. The VNS starts by constructing a solution using a greedy nearest-neighbor algorithm, determining for each SKU in random order the shelf and sequence-position that adds the least distance to the current tour. After that, the VNS iteratively tries to improve the solution by using a *Shaking* operation, destroying parts of the current solution and repairing them in random order, followed by a local search, altering shelves to visit and swapping items between tours. The Python implementation can be found in our GitHub codebase and a detailed description is found in Xie, Li, and Luttmann (2023).

## Results

The results of our experiments comparing different encoder architectures and decoding strategies can be found in Tables 3 and 4, respectively. The HAM encoder clearly outperforms the other tested architectures. While the performance of the MatNet encoder comes close to that of HAM, the other models performed significantly worse with the HAN architecture producing the most inferior results. But also the HGCN is

---

[3]https://github.com/ellelsd/rl4msprp

| Instance | $\|\mathcal{P}\|$ | Gurobi | | | | Heuristic | | | HAM + beam search | | |
|---|---|---|---|---|---|---|---|---|---|---|---|
| | | % opt. | $Z$ | gap % | runtime | $Z$ | gap % | runtime | $Z$ | gap % | run (training) |
| | 3 | 100 | **1.502** | 0.0 | 30s | **1.502** | 0.0 | 3.5s | **1.502** | 0.0 | 0.14s (4h) |
| PRP20 | 6 | 100 | **2.390** | 0.0 | 289s | 2.397 | 0.3 | 5.3s | **2.390** | 0.0 | 0.16s (6h) |
| | 9 | 100 | **2.928** | 0.0 | 36s | 2.945 | 0.6 | 3.8s | 2.929 | 0.0 | 0.19s (7h) |
| | 12 | 40 | 3.639 | 3.9 | 2431s | 3.586 | 2.4 | 67.7s | **3.503** | 0.0 | 0.33s (14h) |
| PRP50 | 15 | 10 | 4.138 | 10.4 | 3264s | 3.852 | 2.8 | 33.3s | **3.748** | 0.0 | 0.35s (15h) |
| | 18 | 0 | 4.030 | 6.2 | 3600s | 3.848 | 1.4 | 65.2s | **3.795** | 0.0 | 0.40s (17h) |
| PRP100 | 15 | 0 | 4.517 | 24.0 | 3600s | 3.682 | 1.1 | 237s | **3.642** | 0.0 | 0.55s (21h) |
| | 20 | 0 | 5.001 | 22.3 | 3600s | 4.154 | 1.6 | 305s | **4.090** | 0.0 | 0.60s (27h) |

Table 5: Comparison of our Heterogeneous Attention Model (HAM) with Gurobi and a heuristic. The gap is w.r.t. the best objective $Z$ across all methods and '% opt.' specifies the percentage of instances that were solved to optimality by Gurobi.

not able to significantly outperform the homogeneous graph AM, despite exploiting the heterogeneous structure of the problem. These results validate the superiority of the multi-head attention mechanism over other neural network architectures on the one hand side, and the effectiveness of the heterogeneous graph formulation on the other hand.

Regarding the decoder part, the hierarchical decoding approach outperforms the other methods tested. The composite decoder, sampling actions from the combinatorial action space, performs the worst. This is in line with Song et al. (2022), who also report large optimality gaps of up to 16%. Furthermore, the performance discrepancy compared to the other methods increases with problem size, suggesting that the exponential growth of the action space leads to problems such as sample inefficiency on larger instances. While the hybrid approach achieves much better results than the composite policy network, the disparity with the hierarchical approach becomes as large as 2.5% for PRP50 instances with 15 SKUs, which clearly outweighs the computational cost of the additional neural decoder.

The comparison of HAM with Gurobi and the heuristic can be found in Table 5. The Gurobi solver can only find optimal solutions for all test instances on PRP20 instances. Remarkably, even on these instances, our HAM combined with beam search achieves on par solution quality with Gurobi. On the larger instances, where Gurobi could not find optimal solutions within the time limit, HAM consistently outperforms both the heuristic and Gurobi. Further, the short inference time allows for fast adaptation of current plans to the dynamics of the warehouse environment.

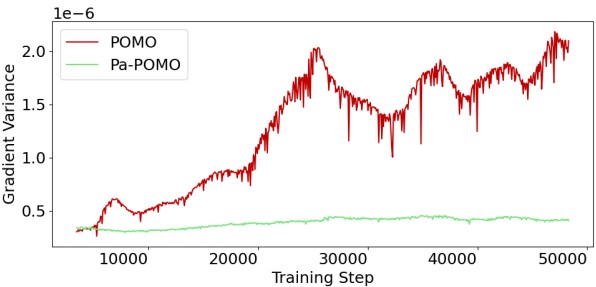

Figure 3: Variance of Gradient estimates

## Conclusion & Discussion

In this work we have introduced an encoder-decoder model to solve CO problems that are defined over heterogeneous graphs and exhibit a combinatorial action space. We believe that our proposed methods can leverage the adoption of NCO for more complex CO problems and pose a serious competitor for handcrafted heuristics. Not only has the hand-crafted heuristic been beaten on the majority of instances, but also the hybrid approach, combining a neural agent with a heuristic, has been proven to be less effective than the end-to-end ML model. Not only is the development of an additional heuristic an extra-burden that NCO tries to eliminate, also is the use of heuristics limited due to the extraordinary efficiency requirement during model training. This usually requires the use of local and greedy search algorithms using only local information, since more sophisticated heuristics would introduce an unaffordable bottleneck during training. Conversely, the neural agent is not only capable to incorporate local information (such as determining which SKU of a given shelf can satisfy the most demand) but also the broader global context (e.g. identifying alternative sources for a particular SKU) in its decision-making. This alone motivates deeper research into DRL-based CO solvers. The notable achievement of our model, surpassing both exact and heuristic solvers on the majority of tested MSPRP instances, further underscores their substantial potential.

One limitation of our work is that the HAM encoder is limited to heterogeneous graphs with only two node types and a single edge type between them. In future work, we will therefore adapt the proposed methods to CO problems with more complex heterogeneous graph representations, e.g. by relaxing the split-order assumption of our MSPRP, which would add another order dimension to the problem. Another important direction for future research is the scalability of ML models to larger instances. Here, our hierarchical decoding strategy makes an important contribution, since it offers the possibility to factorize the action space. Through the factorization, the number of possible actions in our MSPRP grows only linearly with the problem size as opposed to the exponential growth seen in the case of the combinatorial space. In future research we will further investigate means to make the model generalize to larger instances, for instance by training a *foundation model* on instances of different sizes and then fine-tune it to a specific instance type.

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
