# OpenReview forum: "Neural Combinatorial Optimization on Heterogeneous Graphs. An Application to the Picker Routing Problem in Mixed-shelves Warehouses"
_icaps-conference.org/ICAPS/2024/Conference — ICAPS 2024_

### Official Review · Reviewer_UZR6 · 2024-01-19

**Significance And Importance:** 2
**Soundness:** 3
**Novelty:** 2
**Clarity:** 3
**Overall Evaluation:** 2
**Confidence:** 4

**Weaknesses:**

0: Minor weaknesses requiring some work to be addressed for the paper to be accepted.

**Contributions Of The Paper:**

The authors present an MDP representation of the MSPRP and an RL-based framework to provide near-optimal solutions for different problem sizes. In doing so, they promise to provide a solution suitable for more complex real-world problems besides MSPRP. Another contribution is the action space factorisation using a hierarchical decoding strategy.

**Ethical Considerations:**

(1) Not Applicable: The paper does not have any ethical considerations to address

**Nomination For Best Paper:**

No

**Questions For Authors:**

(1) Why did you select MSPRP? What are the benefits regarding the field of NCO?
(2) There are several contributions in the field of NCO that deal with similar complex optimization problems (e.g. JSSP derivates). Why are these problems neglected / not mentioned?
(3) What are other complex optimization problems that benefit from your work? Why?
(4) The reward is defined in a sparse way? How do you deal with invalid sequences? Have you tried different reward designs?

**Reproducibility:**

5: Code and domains (whichever apply) are already publicly available

**Strengths Of The Paper:**

(1) Selection of a not yet covered problem domain (MDP representation for MSPRP)
(2) Contribution to the solution generation of MSPRP using RL and the field of NCO
(3) The authors provide their code and their test data and show convincing results with their approach

**Weaknesses Of The Paper:**

(1) The authors promise to provide a solution suitable for more complex real-world problems, but do not provide any proof regarding the transferability of their approach.
(2) The description of the transition is not in correspondence to the state description
(3) Reward definition seems to be incomplete

---

> ### Author Rebuttal · Authors · 2024-01-27
>
> Thank you very much for reviewing our paper. Please let us address your questions below:
>
> 1. The fast runtimes of NCO algorithms, allowing quick adaptation to environmental changes, are especially beneficial in highly dynamic environments like warehouses. However, NCO literature does not deal with warehousing problems yet, despite their practical relevance. Proving the applicability to such problems could further promote the adoption of NCO.
> Also, we believe the MSPRP to be an illustrative example for the methods we covered. That is, the MSPRP can naturally be formulated as a heterogeneous graph and the action space decomposition is straightforward.
>
>
> 2. You are right that other problems with similar structures exist and have also been covered in NCO literature. Yet, we were eager to apply NCO to a novel problem (also due to the points above).
> We also compared our model with those of other papers who face a similar problem structure (e.g. [3]). Also, we neglected some work like [4] who represent the state of JSSP as a disjunctive graph since it is questionable whether other problems could be formulated as such.
>
> 3. Problems that can be represented as a heterogeneous graph like the JSSP can benefit from our HAM encoder. And the hierarchical decoder is applicable to problems with decomposable action spaces. We demonstrated its superiority in comparison to combined action space approaches, which have been applied to the VRP with a heterogeneous fleet [2] and the JSSP [3] for example.
>
> 4. Yes, the reward function is defined such that the agent observes the ultimate reward only once the solution is complete. This aligns with common NCO practices, as seen in [1]. We did not experiment extensively with other reward designs due to the effectiveness of this approach.
> To handle invalid sequences, we implement a masking procedure at every decoding stage, preventing the selection of actions leading to infeasible solutions.
>
> We highly appreciate your time to review our work and point out its weaknesses. We will change the camera-ready version of our paper accordingly.
>
>
> [1] Kool et al. (2018). Attention, learn to solve routing problems!
>
> [2] Li et al. (2021). Deep reinforcement learning for solving the heterogeneous capacitated vehicle routing problem.
>
> [3] Song et al. (2022). Flexible job-shop scheduling via graph neural network and deep reinforcement learning.
>
> [4] Zhang et al. (2020). Learning to dispatch for job shop scheduling via deep reinforcement learning.

---

### Official Review · Reviewer_zVC7 · 2024-01-22

**Significance And Importance:** 2
**Soundness:** 4
**Novelty:** 2
**Clarity:** 3
**Overall Evaluation:** 2
**Confidence:** 4

**Weaknesses:**

1: Minor weaknesses that are easily fixable.

**Contributions Of The Paper:**

The paper presents a novel encoder-decoder neural network architecture that, combined with reinforcement learning, allows to solve combinatorial optimization problems where the state can be represented as heterogeneous graphs. After providing a detailed technical explanation of several state-of-the-art architectures for neural combinatorial optimization and learning on heterogeneous graphs, the authors introduce their approach. The encoder uses self-attention, while combining linear transformations to align the representation space of the heterogeneous nodes in the graph, while also considering edge features in the process. Then, to overcome the combinatorial aspect of the problems at hand, the decoder makes use of a hierarchical approach. The proposed approach is applied to the picker routing problem in mixed-shelves warehouses, outperforming both heuristics and mathematical optimization in terms of objective and runtime. Additional experiments are provided, showing specific improvements of the proposed encoder and decoder architectures against previous techniques.

An attention-based architecture able to deal with edge features for heterogeneous graphs was introduced in “Multi-Agent Trajectory Prediction With Heterogeneous Edge-Enhanced Graph Attention Network”, by Mo et al. (2022). This might be an interesting comparison for the encoder architecture, since it should be able to deal with multiple types of nodes and connections.

**Ethical Considerations:**

(1) Not Applicable: The paper does not have any ethical considerations to address

**Nomination For Best Paper:**

No

**Questions For Authors:**

1. It is unclear how difficult the problem is. In Table 5, Gurobi is run up to 1 hour, and produces quite large average gap for PRP100. I understand it is not feasible to run Gurobi longer for so many instances. However, it would be interesting to let it run 20 hours for a few instances, and to get some idea on the difficulty of the problem.
2. Is the dummy node connected to all the other nodes? And what are the edge features, if there are any?

**Reproducibility:**

5: Code and domains (whichever apply) are already publicly available

**Strengths Of The Paper:**

.•	Clear exposition of the weaknesses of different approaches.
•	Good experimental results.
•	Very fast runtime, even for large instances. This allows to quickly adapt plans to the highly dynamic warehouse environment

**Weaknesses Of The Paper:**

•	The work only considers routing of one picker while, in real warehouses, multiple pickers need to be coordinated, which is usually more complicated than optimizing their routes individually.
•	The description of how routing instances are generated is not very clear.

---

> ### Author Rebuttal · Authors · 2024-01-28
>
> Thank you very much for your time to review and evaluate our work. Please let us address your questions below:
>
>
> 1. The MSPRP is NP-hard as has been proved by [1], therefore it is quite natural that Gurobi performs badly on the larger instances. However, we agree with you that these results alone hardly give an indication about the overall quality of our model. Nevertheless, we deemed it valuable to include these results as we could obtain (proven) optimal results on the smaller instances using Gurobi, allowing us to demonstrate that our model is capable of finding these solutions as well. For the larger instances we can still compare with the results of the heuristic. The authors of the heuristic in [2] compared it to solutions obtained via Gurobi after 4 hours of runtime and still report very good results. Also, in response to the reviews we ran Gurobi again on the 20 test examples of our largest instance type (PRP100 with 20 SKUs) for 10 hours, but found the gap to decrease only marginally (from 22% to 18%). Lastly, we argue that by comparing our model with other approaches from the NCO literature we provide sufficient evidence for the quality of the developed methods.
>
> 2. The dummy SKU node is not connected to any other node. Instead, we introduce the dummy node embedding, which is a zero vector of equal dimension as the other node embeddings, after the encoder. By choosing a zero vector, the dummy node does not affect the other nodes through the attention mechanism of the decoder. The dummy SKU node is always masked (has a zero sampling probability) except for the case that the agent chooses to return to the depot. Then it is the only possible action.
>
> 3. Edge features only exist between shelf and SKU nodes and represent the remaining supply of a given SKU in the respective shelf. Edge feature between SKU nodes or shelf nodes do not exist.
>
> We highly appreciate your review for pointing out that the paper was not really precise in this regard, and we will change the paper for the camera-ready version accordingly.
>
> [1] Weidinger et al. (2019). Picker Routing in the Mixed-Shelves Warehouses of e-Commerce Retailers.
>
> [2] Xie et al. (2023). Formulating and solving integrated order batching and routing in multi-depot AGV-assisted mixed-shelves warehouses.

---

### Official Review · Reviewer_jW3t · 2024-01-23

**Significance And Importance:** 2
**Soundness:** 3
**Novelty:** 3
**Clarity:** 3
**Overall Evaluation:** 1
**Confidence:** 4

**Weaknesses:**

0: Minor weaknesses requiring some work to be addressed for the paper to be accepted.

**Contributions Of The Paper:**

The paper presents a novel approach to solving combinatorial optimization problems in mixed-shelves warehouses, focusing on the picker routing problem. It introduces a new encoder-decoder model tailored for heterogeneous graphs, which efficiently handles complex combinatorial action spaces. This method demonstrates superior performance compared to traditional algorithms and other neural network architectures. The authors validate their model against established heuristics and exact solvers, proving its effectiveness in practical scenarios. This work not only contributes to neural combinatorial optimization but also shows promise for broader application in more complex combinatorial problems.

**Ethical Considerations:**

(1) Not Applicable: The paper does not have any ethical considerations to address

**Nomination For Best Paper:**

No

**Questions For Authors:**

If possible, please provide responses to the comments made in the "Weaknesses" box.

**Reproducibility:**

5: Code and domains (whichever apply) are already publicly available

**Strengths Of The Paper:**

- Formulate the Mixed-shelves Picker Routing Problem (MSPRP) for mathematical programming approaches and neural combinatorial optimization (NCO) solver.

- Introduce an encoder-decoder architecture that handles factored action space hierarchically. With structure exploited, this framework is superior to most existing NCO methodologies.

- The application domain is more complex and interesting than most existing NCO domains.

- Code and implementation are already made available anonymously.

**Weaknesses Of The Paper:**

1. The presented mathematical programming formulations are not optimized. Comparing HAM against them is thus not the most informative.

2. While the running time of the HAM approach looks fast, the training times are very long and do not seem to scale well in response to the problem size.

3. The largest instance size is with 100 storage locations. For a realistic warehouse, the storage locations might be several orders of magnitude larger. With the reported training times, will this NCO approach be applicable to real-world instances?

4. As the HAM models are pre-trained, what if there are minor changes to the problem instance configuration (e.g., in terms of the number of storage locations, and SKU-related information); could the trained models be used to handle instances that are slightly different in sizes/state dimensions?

---

> ### Author Rebuttal · Authors · 2024-01-28
>
> Thank you very much for your time to review our work. Please let us address your concerns below:
>
> 1. We agree with you that the results obtained by Gurobi alone are a weak indicator of the quality of our model. However, we have included the Gurobi model because it provides us with optimal solutions on the smaller instances, and we have been able to show that our model is also able to find these solutions. Moreover, we believe that by also comparing the model with a strong heuristic and other NCO approaches, we provide sufficient evidence for the quality of the developed methods.
>
> 2. You are correct that the training time scales poorly with the size of the instances. The self-attention mechanism is known to scale quadratically with the size of the sequence. Efficient transformer implementations are a hot-topic in ML research right now, but lay beyond our scope. We included some possible approaches to scale to larger instances below.
>
> 3. You are correct that real-world instances are much larger than the ones we considered in our evaluation. Solving instances of practical size is an actively addressed research question in the NCO field (see e.g. [1,2]) and we believe that by showing promising results with the decomposition of the action space, effectively reducing its size, we have contributed to this regard.
> One obvious approach to scale to larger instances is to parallelize over multiple GPUs.
> A more elaborated approach is provided by [1], who propose a knowledge-distillation scheme for the NCO domain. In this framework, our proposed architecture could be employed as a teacher model to “teach” a lightweight student model, which in turn could be used for larger instances.
>
> 4. The model architecture is size-agnostic due to the attention mechanisms and thus a trained model can be transferred to instances of any size. However, making NCO models generalize to instances of different sizes is still an open research gap (also see [1] for example), which lies outside the scope of our paper. Therefore, we tested models on instances of the same size they were trained on.
>
> We highly appreciate your time to review our work and point out these weaknesses. We will change the camera-ready version of our paper accordingly.
>
> [1] Bi et al. (2022). Learning generalizable models for vehicle routing problems via knowledge distillation
>
> [2] Luo et al. (2023). Neural combinatorial optimization with heavy decoder: Toward large scale generalization

---

### Meta-Review · Area_Chair_nHD6 · 2024-02-02

**Recommendation:** Accept (Oral)
**Confidence:** 5

**Metareview:**

The paper introduces an NCO method for the Mixed-shelves Picker Routing Problem. Compared to existing NCO methods, this paper addresses a novel and compelling application domain and proposes a new NCO architecture. The rebuttal has effectively addressed most concerns raised by the reviewers. We recommend that the authors improve the final version based on the comments from the reviewers if the paper gets accepted.

**Ethical Considerations:**

(1) Not Applicable: The paper does not have any ethical considerations to address